# A Novel CT Perfusion-Based Fractional Flow Reserve Algorithm for Detecting Coronary Artery Disease

**DOI:** 10.3390/jcm12062154

**Published:** 2023-03-09

**Authors:** Xuelian Gao, Rui Wang, Zhonghua Sun, Hongkai Zhang, Kairui Bo, Xiaofei Xue, Junjie Yang, Lei Xu

**Affiliations:** 1Department of Radiology, Beijing Anzhen Hospital, Capital Medical University, Beijing 100029, China; 2Discipline of Medical Radiation Science, Curtin Medical School, Curtin University, Perth 6845, Australia; 3School of Biomedical Engineering, Sun Yat-sen University, Shenzhen 518107, China; 4Department of Cardiology, The Sixth Medical Center, Chinese PLA General Hospital, Beijing 100048, China

**Keywords:** fractional flow reserve from CT angiography, computed tomographic perfusion, fractional flow reserve, coronary artery disease, myocardial ischemia

## Abstract

Background: The diagnostic accuracy of fractional flow reserve (FFR) derived from coronary computed tomography angiography (CCTA) (FFR-CT) needs to be further improved despite promising results available in the literature. While an innovative myocardial computed tomographic perfusion (CTP)-derived fractional flow reserve (CTP-FFR) model has been initially established, the feasibility of CTP-FFR to detect coronary artery ischemia in patients with suspected coronary artery disease (CAD) has not been proven. Methods: This retrospective study included 93 patients (a total of 103 vessels) who received CCTA and CTP for suspected CAD. Invasive coronary angiography (ICA) was performed within 2 weeks after CCTA and CTP. CTP-FFR, CCTA (stenosis ≥ 50% and ≥70%), ICA, FFR-CT and CTP were assessed by independent laboratory experts. The diagnostic ability of the CTP-FFR grouped by quantitative coronary angiography (QCA) in mild (30–49%), moderate (50–69%) and severe stenosis (≥70%) was calculated. The effect of calcification of lesions, grouped by FFR on CTP-FFR measurements, was also assessed. Results: On the basis of per-vessel level, the AUCs for CTP-FFR, CTP, FFR-CT and CCTA were 0.953, 0.876, 0.873 and 0.830, respectively (all *p* < 0.001). The sensitivity, specificity, accuracy, positive predictive value (PPV) and negative predictive value (NPV) of CTP-FFR for per-vessel level were 0.87, 0.88, 0.87, 0.85 and 0.89 respectively, compared with 0.87, 0.54, 0.69, 0.61, 0.83 and 0.75, 0.73, 0.74, 0.70, 0.77 for CCTA ≥ 50% and ≥70% stenosis, respectively. On the basis of per-vessel analysis, CTP-FFR had higher specificity, accuracy and AUC compared with CCTA and also higher AUC compared with FFR-CT or CTP (all *p* < 0.05). The sensitivity and accuracy of CTP-FFR + CTP + FFR-CT were also improved over FFR-CT alone (both *p* < 0.05). It also had improved specificity compared with FFR-CT or CTP alone (*p* < 0.01). A strong correlation between CTP-FFR and invasive FFR values was found on per-vessel analysis (Pearson’s correlation coefficient 0.89). The specificity of CTP-FFR was higher in the severe calcification group than in the low calcification group (*p* < 0.001). Conclusions: A novel CTP-FFR model has promising value to detect myocardial ischemia in CAD, particularly in mild-to-moderate stenotic lesions.

## 1. Introduction

Coronary computed tomography angiography (CCTA) can provide excellent anatomical information on coronary arteries and has a high negative predictive value for detecting obstructive coronary artery disease (CAD) [1,2,3]. Although CCTA is widely used for clinical assessment of CAD, it does not provide functional assessment of coronary arteries [4,5]. Fractional flow reserve (FFR) derived from coronary computed tomography angiography (FFR-CT) is becoming a gatekeeper for assessing hemodynamics of coronary arteries and the guide for revascularization, with better agreement with fractional flow reserve (FFR) [6,7,8].

Traditional FFR-CT based on computational fluid dynamics (CFD) and deep learning relies on a population-averaged physiological hypothesis model to estimate exit resistance to simulate boundary conditions. However, due to individual differences between patients, this quantification and distribution of total coronary blood flow can affect coronary exit resistance and therefore the accuracy of FFR-CT [9,10]. Thus, FFR-CT needs to be further improved regarding the diagnostic accuracy and the identification of the gray zone (FFR between 0.75 and 0.80) [11].

Previous studies have demonstrated that the equivalent circuit model of coronary circulation is mainly influenced by the outlet resistance and that the effect of different values for the inlet pressure on the FFR can be entirely ignored [12,13]. A study has developed a digital arterial model by improving the outlet resistance of the artery, providing important information on the magnitude of the hemodynamic parameters experienced in localized sites. This provides hope for future studies addressing medical implant design [14]. In addition, parameters such as outlet resistance can predict the formation and progression of plaque [15]. It has also been shown that the accuracy of hemodynamics in the abdominal aorta and visceral arteries can be improved by adjusting relevant parameters such as outlet resistance [16]. Therefore, focusing on outlet resistance to improve boundary conditions may lead to the discovery of more experimental work on the cardiovascular system, and even treatment or prognosis in the future.

Dynamic stress myocardial computed tomography perfusion (CTP) has been shown to be accurate for quantitative and semi-quantitative assessment of myocardial ischemia in single and multicenter studies [17,18,19]. CTP was progressively optimized to reduce radiation doses, and the CTP-myocardial blood flow (*MBF*) was comparable to FFR-CT in identifying the lesions causing ischemia [20]. Currently, myocardial CTP-based FFR (CTP-FFR) models have been proposed [21]. CTP is performed during pharmacological congestion, which is the same physiological state as when invasive FFR is measured. Thus, *MBF* within the myocardial perfusion territory can directly quantify the total coronary blood flow better simulating the boundary conditions of FFR-CT. This approach has been validated in our previous study showing the repeatability and accuracy of using CTP-FFR in assessing coronary stenosis [21].

Thus, we developed a new algorithm based on the CTP to measure FFR through outlining myocardial perfusion regions and improving boundary conditions. The aim of this study was to investigate the feasibility and diagnostic performance of this new CTP-FFR in detecting ischemia while using invasive FFR as the reference standard.

## 2. Materials and Methods

### 2.1. Study Design and Patients

The study was designed to use the CTP-based outlet boundary condition (BC) model to calculate CTP-FFR in patients with suspected CAD.

This research was approved by the local clinical institutional ethics committee. All the patients provided written informed consent. A total of 93 patients (37 patients from Beijing Anzhen Hospital, Capital Medical University, and 56 patients from the First Medical Centre of the General Hospital of the Chinese People’s Liberation Army) with suspected CAD were finally enrolled in this retrospective study from March 2019 to September 2020. Among all the participants, stress CTP and CCTA were performed first, followed by invasive coronary angiography (ICA), quantitative coronary angiography (QCA) and invasive FFR measurements, which were performed within two weeks. Patients with a QCA ranging from 30 to 90% luminal narrowing were included in this study. The study flow chart is shown in Figure 1. Exclusion criteria were as follows: (1) contraindications to the adenosine triphosphate (ATP) stress tests and iodinated contrast agents; (2) women who were pregnant; (3) patients decompensating heart failure and severe liver and renal insufficiency (estimated glomerular filtration rate < 60 mL min^−1^ 1.73 m^−2^); (4) history of revascularization or stent; (5) acute coronary syndrome; and (6) CCTA and CTP images could not be evaluated due to suboptimal image quality.

### 2.2. CCTA and CTP Acquisition

Patients were asked to refrain from smoking and caffeine for 24 h and fast for 6 h before the scan. The stress CTP and CCTA were performed using two type of CT vendors, 256-row CT scanner (Revolution CT, GE Healthcare, Milwaukee, WI, USA); 128-row dual-source CT (Somatom Definition Flash, Siemens Healthineers, Forchheim, Germany) cardiac CT protocol consisted of: (1) a positioning scan; (2) a CT calcium score scan; (3) a dynamic stress CTP; and (4) a coronary CTA according to the recommendations of the Society of Cardiovascular Computed Tomography (SCCT). Stress CTP scans were performed during an intravenous injection of ATP (160 μg/kg/min for 3 min). Scanning parameters were as follows: z-coverage, 14 cm; matrix size, 512 × 512; voxel size, 0.625 mm; thickness, 1.25 mm; gantry rotation time, 0.28 s; and tube voltage, 80 kV; for tube current, the Smart-mA technique with 200 mA was used. The exposure time of the stress CTP was 30 s. The contrast agent (Ultravist, 370 mg/mL of iopromide; Bayer, Wayne, NJ, USA) was intravenously injected in the right antecubital vein at a flow rate of 4 to 5 mL/s. Data acquisition was triggered using a bolus-tracking technique, and 50 to 60 mL of the contrast agent was injected, followed by 30 to 35 mL of 0.9% saline solution. A single CCTA scan with the same parameters was performed after 20 min of CTP. The scan range covered the entire heart, from the trachea bifurcation up to the diaphragm. Image data were transferred to a workstation (Siemens Syngo. via, Erlangen, Germany) for quantitative analysis of dynamic CTP including MBF, time attenuation curve (TAC) and left ventricular mass (LVMASS). From the *MBF* map, we selected the one that best represented the area of myocardium associated with the target vessel. Within this cross-section, an area of interest with a minimum area of 50 mm^2^ was outlined to sample the *MBF* within the suspected perfusion defect [22]. The total coronary artery calcium score (CACS) was measured.

The effective radiation dose was calculated by multiplying the dose–length product by a constant coefficient (k = 0.014 mSvmGy^−1^cm^−1^) [23].

### 2.3. The FFR Based on CTP

The CTP-FFR model and algorithm has recently been developed [21]. Evaluation of CTP-FFR was divided into the following steps (a detailed flow chart is shown in Figure 2). First, CCTA images were segmented by a semi-automatic segmentation algorithm for coronary artery 3D reconstruction and truncation. Next, for CFD simulation and FFR-CT computation, we used the open-source software Open FOAM for numerical simulations of fluid dynamics. The numerical simulation was governed by the following momentum and mass conservation equations for an incompressible fluid.

Finally, with regards to the inlet and outlet boundary conditions, the inlet boundary condition employed the mean aortic pressure (*MAP*), which can be calculated according to the brachial cuff-based pressure, as *MAP* = 0.4 × (SBP − DBP) + DBP, where SBP and DBP are brachial systolic and diastolic blood pressures, respectively. With regards to the outlet boundary conditions, a previous study has shown that the *MBF* calculated by the volume perfusion CT software is greatly underestimated compared with the stress *MBF* quantified by the gold standard positron emission tomography (PET) [24]. Ishida et al. proposed to calculate the hematocrit–myocardial transfer constant *K*1 to replace the *MBF* of volume perfusion CT [25]. In addition, Kikuchi et al. [26] employed a method to correct *K*1 to *MBF*. The Renkin–Crone formula is expressed between *K*1 and *MBF*, as
(1)$$\begin{array}{l} K1=\left[1-0.904\exp\left(-1.203/MBF\right)\right]MBF \end{array}$$

In this study, *MBF* was corrected by using this formula. We used the Voronoi algorithm to distribute myocardial perfusion [27], which has been demonstrated to assess stenosis-specific myocardial perfusion territories accurately. Therefore, the coronary blood flow at the *i*-th outlet ($Q_{out,i}^{hyp}$) is calculated as
(2)$$Q_{out,i}^{hyp}=\sum_{n=1}^{N}Vol_{n}\times MBF_{n}$$
where $Vol_{n}$ is the single voxel volume in myocardial perfusion territory, N is the number of all voxels in the myocardial perfusion territory, and $MBF_{n}$ is the *MBF* in a single voxel. The resistance of the coronary branch at the *i*-th outlet ($R_{i}^{hyp}$) is expressed as
(3)$$R_{i}^{hyp}=\frac{MAP-P_{v}}{Q_{out,i}^{hyp}}$$
where MAP is the mean aortic pressure, and $P_{v}$ is the venous pressure in the venous vessels and is set to 5 mmHg.

**Figure 2 jcm-12-02154-f002:**
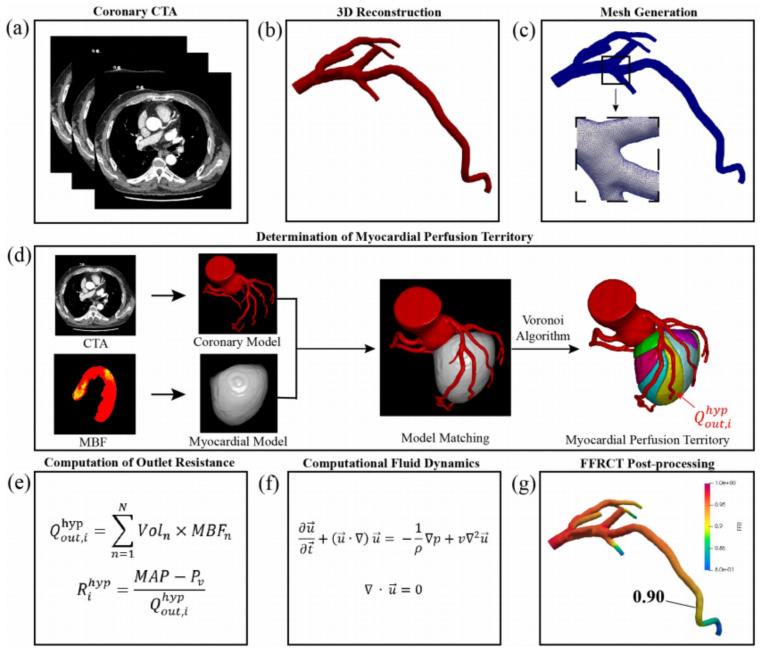
The process of FFR non-invasive calculation based on CTP. (**a**) Acquisition of coronary CTA; (**b**) 3D reconstruction of coronary artery; (**c**) meshing of 3D models of coronary artery; (**d**) determination of myocardial perfusion territory; (**e**) computation of outlet resistance; (**f**) Navier–Stokes equations that govern the fluid dynamics of blood; (**g**) post-processing of patient-specific FFR-CT. Adapted from Xue et al. [21].

### 2.4. The Interpretation of CCTA, CTP, FFR-CT

In all patients, an observer (X.X) who had five years’ experience in interpreting cardiovascular disease was blinded to invasive FFR and clinical data, and marked and calculated the stress CTP and CCTA. FFR-CT measurements were performed by an independent core laboratory at Keya Medical in a manner that was blinded to the clinical findings [28].

### 2.5. Invasive Coronary Angiography and FFR

Conventional multi-position selective angiography via radial artery or femoral artery was performed by cardiovascular imaging machine (GE Innova3100, The Netherlands). QCA and invasive FFR measurements were performed on the vascular branches with stenosis in angiography. Then, ATP (140–180 μg/kg/min) was injected into the peripheral vein to induce the coronary artery to reach the maximum congestion state, and the FFR value was read by the pressure guide wire of a FFR measuring system (ST. Jude Medical Co., Ltd., Shanghai, China). The pressure wire was positioned in a vessel segment (≥1.5 mm in diameter) that was 20 mm distal to a stenosis. QCA and FFR were analyzed at the clinical site by a cardiologist with five years of experience who was blinded to stress CTP and CCTA findings.

### 2.6. Statistical Analysis

Statistical analysis was performed with SPSS version 25.0 (IBM SPSS Statistics, IBM Corporation, Armonk, NY, USA) and MedCalc (v20.1.0). Quantitative variables were expressed as mean ± SD if normally distributed, while those that did not conform to the normal distribution were represented by the median and inter quartile range (IQR). Categorical variables were expressed as frequency and rate. The sensitivity, specificity, accuracy, positive predictive value (PPV) and negative predictive value (NPV), and their corresponding 95% confidence intervals (CIs) were calculated for CTP-FFR, CTP, FFR-CT and the combination of the three for the diagnosis of ischemic lesions. In addition, diagnostic performance of CCTA with stenosis ≥ 50% and ≥ 70% was calculated. The Chi-square test and the McNemar test were used to compare sensitivities, specificities and accuracies to compare diagnostic performance characteristics in subgroups, and the DeLong test was used to compare areas under the curve (AUCs) of receiver-operating characteristics among CTP-FFR, CTP, FFR-CT and CCTA. FFR ≤ 0.8 was used as the boundary value for diagnosis of ischemic lesions. Diagnostic performance of CTP-FFR was also compared between subgroups stratified by coronary artery calcium (CAC) scoring < 400 and ≥ 400 [29], stenosis of QCA (mild, 30–49%; moderate, 50–69%; and severe, ≥70%), vessel nomenclature (left anterior descending coronary artery and left circumflex coronary artery) or lesion location on vessel (proximal, middle and distal). The Pearson correlation coefficient and Bland–Altman analysis were used to analyze the correlation between CTP-FFR with invasive FFR. A *p* < 0.05 value was considered statistically significant.

## 3. Results

### 3.1. Patient Characteristics

Baseline characteristics of the study population are shown in Table 1. A total of 93 patients (71 men and 22 women; 59.2 ± 10.9 years old) were finally enrolled and successfully underwent CCTA and the stress CTP, with 103 vessels undergoing the FFR and QCA examination for the analysis in our study, including LAD 82% (84/103) and LCX 18% (19/103). On a per-vessel basis, medians value of 0.81 (IQR: 0.74 to 0.87) and 0.82 (IQR: 0.71 to 0.86) were found for CTP-FFR and invasive FFR, respectively. The detailed patient baseline characteristics are shown in Table 1.

### 3.2. CCTA, CTP and ICA Measurements

Among the 103 vessels, proximal, middle and distal stenoses were found on CCTA in 59 (57%), 38 (37%) and 6 (6%) vessels, respectively. All vessels were divided into 79 (77%) with low calcification (CAC < 400) and 24 (23%) with severe calcification (CAC ≥ 400), according to the Agatston scores (Table 1).

QCA showed coronary stenosis of < 50% in 28 vessels (27%), and stenosis of ≥ 50% in 75 vessels (73%), of which 55 vessels (53%) were ≥ 70% stenosis. Eventually, ICA and FFR demonstrated hemodynamically significant stenosis by an invasive FFR ≤ 0.80 in 47 (46%) out of 103 vessels, and 44 (47%) out of 93 patients (Table 1).

### 3.3. Diagnostic Performance of CCTA, FFR-CT, CTP and CTP-FFR

The AUCs of CCTA for evaluating stenosis of ≥ 50%, CTP and FFR-CT on a per-vessel analysis were 0.830, 0.876 and 0.873, respectively (Figure 3). On a per-vessel analysis for detecting flow-limiting stenosis ≥ 50%, the sensitivity, specificity, accuracy, PPV and NPV for CCTA alone were 0.87, 0.54, 0.69, 0.61 and 0.83, respectively. For ≥ 70% stenosis, the corresponding values on per-vessel analysis were 0.75, 0.73, 0.74, 0.70 and 0.77, respectively (Table 2).

The AUC of CTP-FFR for per vessel was 0.953 (Figure 3), which was significantly higher than CTP, FFR-CT or CCTA interpretation on a per-vessel level (all *p* < 0.05). The diagnostic performance of CTP-FFR ≤ 0.8 for detecting CAD, compared with invasive FFR, is provided in Table 2. CTP-FFR demonstrated a vessel-based sensitivity, specificity, accuracy, PPV and NPV of 0.87, 0.88, 0.87, 0.85 and 0.89, respectively.

On a per-vessel basis, both for ≥ 50% and ≥ 70% stenosis, higher specificity and accuracy of CTP-FFR over CCTA were demonstrated (all *p* < 0.05). In addition, the sensitivity and accuracy of CTP-FFR + CTP + FFR-CT were also improved over FFR-CT alone (both *p* < 0.05). As for specificity, it was also improved compared with FFR-CT or CTP alone (both *p* < 0.01).

On a per-vessel basis, the CTP-FFR had a strong correlation with invasive FFR (Pearson’s correlation coefficient 0.89, 95%CI: 0.85–0.93, *p* < 0.001) (Figure 4). Bland–Altman analysis showed that the mean difference in total vessels was 0 (95% limits of agreement –0.106 to 0.106; Figure 4).

### 3.4. Diagnostic Performance of CTP-FFR in the Subgroups Analysis

CTP-FFR and FFR were in good agreement on a per-vessel basis regardless of calcium score burden: CAC < 400 (mean difference 0.002; 95% limits of agreement: −0.106 to 0.110), CAC ≥ 400 (mean difference −0.007; 95% limits of agreement: −0.105 to 0.090) (Figure 5). The specificity of CTP-FFR was significantly higher in the severe calcification group than in the low calcification group (*p* < 0.001, Table 3). Figure 6 shows a case of CTP-FFR with CCTA and ICA.

Regardless of the total calcium burden, CTP-FFR showed improved diagnostic sensitivity, specificity for vessels with 30–49% and 50–69% stenosis compared with vessels with stenosis ≥ 70% as defined by QCA (all *p* < 0.01, Table 3). CTP-FFR demonstrated comparably high diagnostic performance between subgroups. In short, with the exception of QCA, the presence of these lesion characteristics did not significantly impact the diagnostic performance of CTP-FFR.

### 3.5. Performance of CTP-FFR Grouped by FFR

The diagnostic accuracies of CTP-FFR in determining hemodynamic significance among different groups of FFR ranges were shown in Figure 7. On a per-vessel basis, CTP-FFR correctly identified all 22 vessels with an FFR ≤ 0.70 and all 12 vessels with an FFR > 0.90 by using the standard threshold FFR of 0.80. CTP-FFR correctly identified 19 vessels with an FFR of 0.70 to 0.80, while the other 6 vessels in this group were false negatives. In the subgroup with an FFR of 0.8–0.9, CTP-FFR was correctly judged in 37 out of 44 vessels, 7 of which were false positives. A significant difference was found in the diagnostic accuracy of one group’s (FFR 0.70 to 0.80) lesions versus the other group with FFR ≤ 0.7 (*p* = 0.014). However, no significant difference was found when comparing the accuracy of determining functional significance in patients with the group FFR 0.70 to 0.80 versus lesions with 0.8 < FFR ≤ 0.9 (*p* = 0.409) or versus lesions with FFR > 0.9 (*p* = 0.064).

## 4. Discussion

We proposed a CTP-based outlet BC model involving the quantification of total hyperemic coronary blood flow and the distribution of outlet coronary blood flow to estimate outlet resistance and FFR-CT, and validated the feasibility of CTP-FFR. Our main findings of this study included: (1) CTP-FFR has a higher diagnostic performance for patients with CAD than CCTA or conventional FFR-CT or CTP, and the combination of CTP-FFR + CTP + FFR-CT has specific incremental diagnostic value. (2) For vessels with mild and moderate stenosis, CTP-FFR has higher diagnostic performance. (3) CTP-FFR is less affected by coronary calcification.

Non-invasive FFR-CT provides coronary hemodynamic information and is considered as a gatekeeper and guides clinical decision-making in patients with CAD [8,30]. Previously, FFR-CT modeled CFD primarily by transluminal attenuation gradients (TAG) and diameter and left ventricular mass (LVM) to estimate pressure [31,32,33], and used patient-specific physiological assumptions to model boundary conditions, which are influenced by exit resistance, the reliability of which in turn depends on the physiology of the coronary blood flow circulation hypothetical model and is limited by it. There is currently no definitive physiological assumption that can accurately quantify patient-specific coronary blood flow and outlet resistance [34,35]. Although artificial intelligence (AI)-based FFR-CT (AI-FFR-CT) is maturing and enables timely on-site processing [36], it is still influenced by image quality, severe calcification and the maximum level of congestion simulated by nitroglycerin. In contrast, dynamic CTP is a non-invasive functional imaging method that quantifies *MBF* during pharmacological congestion by means of myocardial enhancement patterns following contrast injection as a means of estimating outlet resistance and assigning areas of myocardial perfusion [37], with a diagnostic value similar to MRI and positron emission tomography [38,39]. In addition, the use of techniques such as AI and machine learning (ML), standard low voltage (80 kv), automatic tube current modulation and iterative reconstruction has reduced reconstruction time and improved the quality of low-dose myocardial CTP images [40,41].

Our study showed that the AUC of CTP-FFR was higher than CCTA, *MBF* or conventional FFR-CT in detecting CAD, and significantly higher than previous AUCs on per-vessel analysis: 0.79–0.92 [6,32,33,42]. CTP-FFR implements an innovative algorithm for FFR-CT, which in combination with *MBF* and FFR-CT significantly improves the diagnostic performance of non-invasive imaging based on CT. Furthermore, Pearson’s and Bland-Altman analyses also showed strong correlation and agreement between CTP-FFR and FFR. This demonstrated the reliability of CTP-FFR and it was consistent with previous studies that stress CTP has a comparable diagnostic value to FFR-CT, and can further improve diagnostic performance in combination with CCTA and FFR-CT [12,43,44]. Li et al. [45] also found that CTP-*MBF* was superior to ML-based FFR-CT in detecting lesions with severe stenosis. Our results further validated their findings.

Another important finding of our study was that CTP-FFR showed improved diagnostic sensitivity and specificity for vessels with 30–49% and 50–69% stenosis compared with vessels with stenosis ≥ 70%, as defined by QCA. As for intermediate coronary stenoses (luminal stenosis 30–70%), the accuracy and specificity of FFR-CT was 80–86% and 83–85%, respectively [6,32,42]. Donnelly et al. prospectively investigated a novel in situ FFR-CT algorithm and reported a specificity of 72% for detecting intermediate lesions [46]. In our study, CTP-FFR also had the highest diagnostic value (>80%) in patients with mild and moderate stenosis of the QCA, and this further confirms the improved diagnostic performance of CTP-FFR when compared with CCTA and other approaches.

The diagnostic accuracy was higher in the FFR ≤ 0.7 group than in lesions with an FFR of 0.70 to 0.80 (*p* = 0.014). This suggests that CTP-FFR may achieve excellent agreement in the presence of severe ischemia. Current studies suggest that, for the diagnostic value of the gray zone, CTP can provide incremental value on top of FFR-CT [22,47]. Due to the limited sample size in this study, further research with inclusion of larger sample sizes is needed to validate the diagnosis of CTP-FFR grouping by FFR, especially in the gray zone in the future.

Finally, we found high concordance for CTP-FFR, even in the presence of severe calcification (CAC ≥ 400). The results showed that CTP-FFR mildly underestimated the severity of local hemodynamics in 103 vessels among the 93 subjects, resulting in 6 false negatives and 7 false positives at 0.7 < FFR ≤ 0.8 and 0.8 < FFR ≤ 0.9, respectively. Eleven of them occurred in the low calcified plaque group with a CAC < 400 and two in the severe calcified plaque group with a CAC ≥ 400. It is suggested that the diagnostic accuracy of CTP-FFR may be less influenced by calcification and more related to the degree of ischemia. Previous studies found in patients with severe calcification, stent placement or coronary artery bypass grafting (CABG) that CTP was superior to FFR-CT in diagnosing ischemia and could guide the outcome after revascularization [48,49]. Our results further confirmed these findings that CTP-FFR is an accurate tool in assessing stenotic coronary lesions, even in the presence of severe calcification.

Based on the non-Newtonian mechanical characteristics of blood and the development of the boundary conditions [50,51], several studies have confirmed that CFD can construct numerical models to obtain the hemodynamic parameters of blood vessels [52,53]. The accuracy of CFD can be significantly improved by sketching and changing the boundary conditions, such as placing multi-layer prism elements [54]. This shows that boundary conditions are currently the focus of optimization for CFD and that improvements in boundary conditions such as CTP-FFR may also be evaluated in the future to guide the development of numerical models and the treatment of targeted drugs that follow the blood circulation [55].

Our study had several limitations. Firstly, this was a retrospective study involving two clinical sites and the results may be somewhat biased due to the small sample size. Subsequently we will also include more subjects and increase the number of gray zones. Secondly, due to the extremely thin myocardium of the right coronary artery, it was challenging to outline the myocardial perfusion zone by an automated algorithm. The *MBF* of the right ventricle was not calculated. Thirdly, the scans of the stress CTP were performed by GE and Siemens vendors, respectively, and there may have been an effect of machine efficacy on the images; however, we adjusted the parameters and scan sequences to be consistent to minimize the potential impact.

## 5. Conclusions

Compared with CCTA or traditional FFR-CT or CTP, a newly developed CTP-FFR model offers higher diagnostic value for patients with CAD, especially in mild and moderate stenosis. It also shows the potential value of CTP-FFR in assessing coronary lesions with severe calcification. This offers new options for patients with coronary artery disease and may further enhance clinical diagnosis and improve treatment in the future.

## Figures and Tables

**Figure 1 jcm-12-02154-f001:**
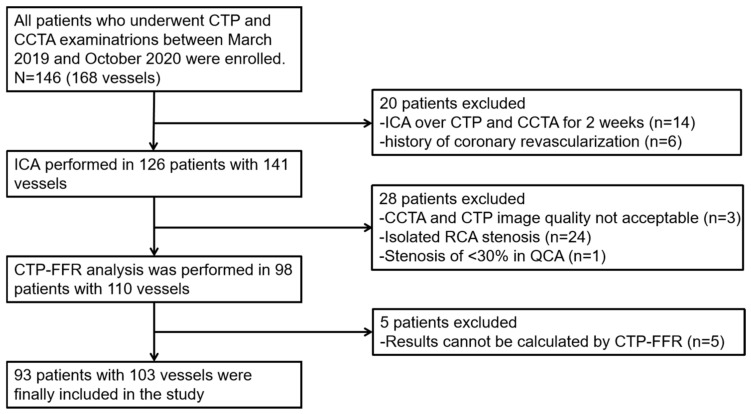
Study flow chart. CCTA = coronary computed tomography angiography; CTP-FFR = computed tomography perfusion (CTP)-derived fractional flow reserve (FFR); ICA = invasive coronary angiography; QCA = quantitative coronary angiography; RCA = right coronary artery.

**Figure 3 jcm-12-02154-f003:**
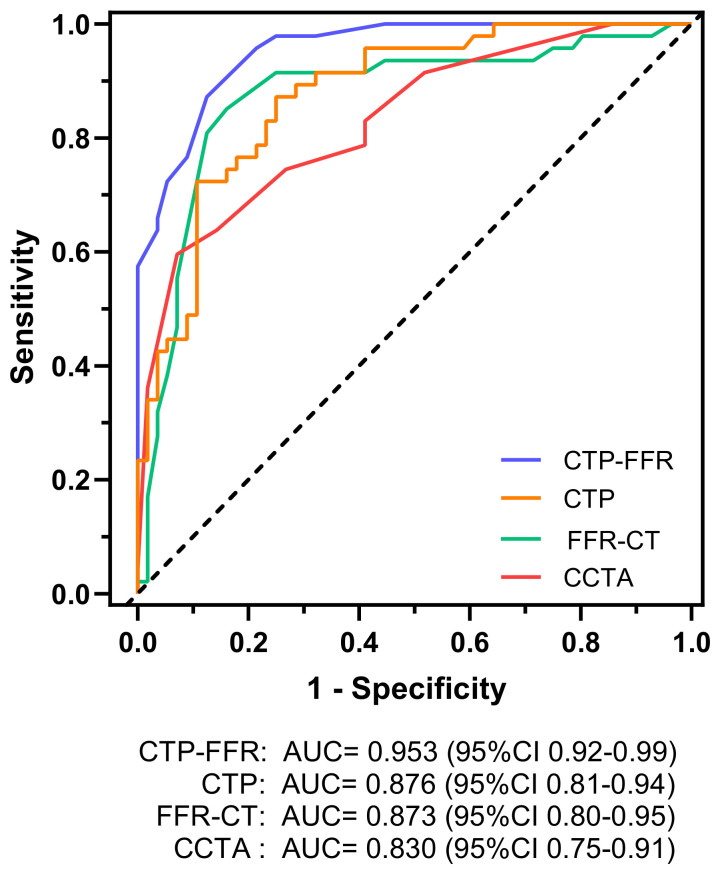
Graph showing diagnostic performance of CTP-FFR, CTP, FFR-CT and CCTA. AUC of receiver operating characteristics curve analysis is shown on per vessel for CTP-FFR, CTP, FFR-CT and visual stenosis grading (stenosis ≥ 50%) at CCTA. The dotted line represents the reference line. AUC = area under receiver operating characteristics curve. Other abbreviations same as in Figure 1.

**Figure 4 jcm-12-02154-f004:**
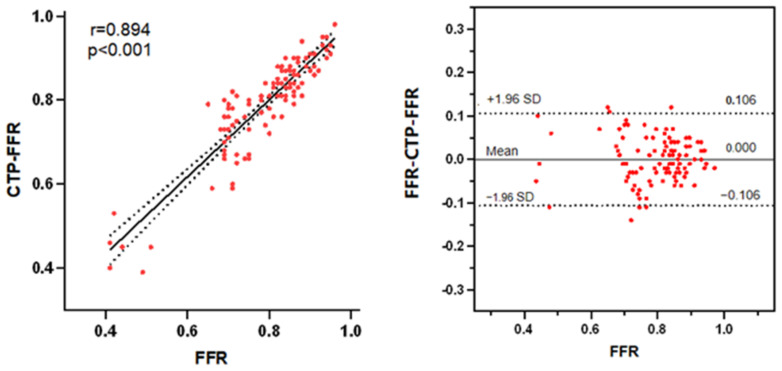
(**Left**) A strong correlation (R = 0.894) is observed between CTP-FFR and FFR. (**Right**) Bland–Altman analysis of FFR and CTP-FFR on a per-vessel basis. The distribution of red points represents the correlation and consistency of CTP-FFR with FFR. Abbreviations same as in Figure 1.

**Figure 5 jcm-12-02154-f005:**
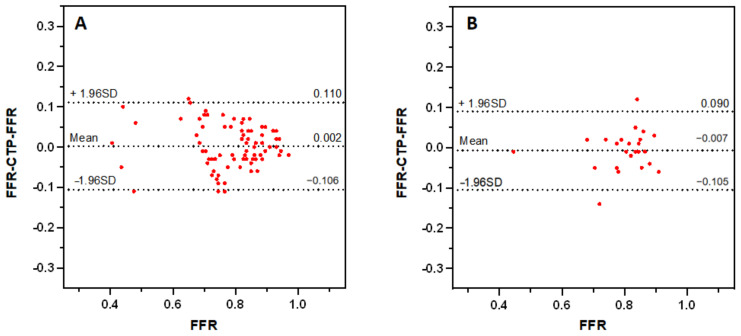
Bland–Altman analysis of FFR and CTP-FFR on a per-vessel basis in low calcification ((**A**), CAC < 400) and severe calcification ((**B**), CAC ≥ 400), respectively. The distribution of red points represents the consistency of CTP-FFR with FFR. CAC = coronary artery calcium. Other abbreviations as in Figure 1.

**Figure 6 jcm-12-02154-f006:**
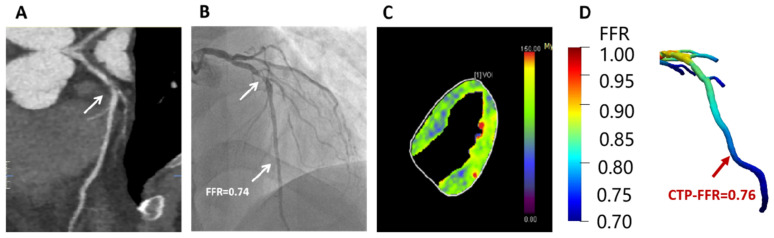
Example of a 47-year-old man who presented with atypical chest pain, hypertension and dyslipidemia. (**A**) CCTA shows stenosis (arrow) is caused by a mixed plaque of the proximal LAD. (**B**) ICA shows severe stenosis (arrow) with ischemia in LAD (FFR = 0.74). (**C**) Stress-CTP shows myocardial blood flow (**D**) CTP-FFR demonstrates value of 0.76 in the LAD. Abbreviations same as in Figure 1 and Table 1.

**Figure 7 jcm-12-02154-f007:**
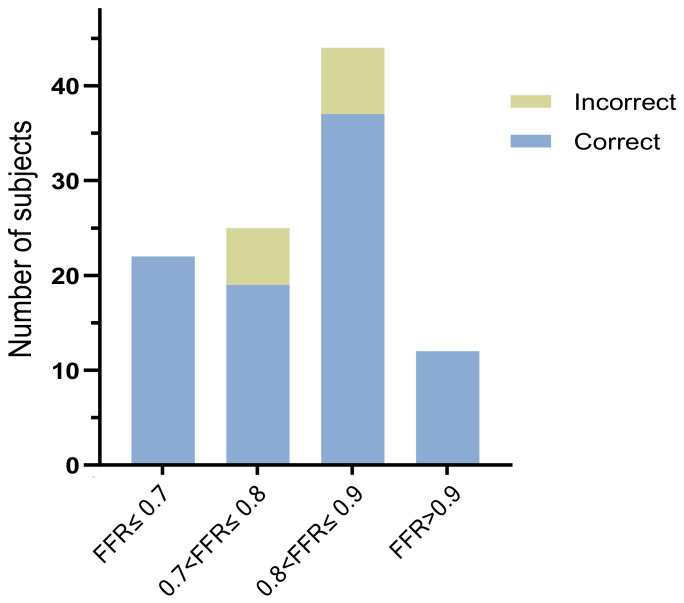
The diagnostic accuracy of CTP-FFR in correctly identifying the hemodynamically significant lesions among the different invasive FFR groups. All abbreviations same as in Table 1.

**Table 1 jcm-12-02154-t001:** Baseline characteristics of the study population.

Clinical Characteristics	N = 93
Age, y	59.2 ± 10.9
Male	71 (76.3)
BMI, kg/m^2^	25.89 ± 2.48
Hyperlipidemia	50 (53.8)
Hypertension	51 (54.8)
Diabetes	24 (25.8)
Current smoker	37 (39.8)
**Symptoms**	
Typical angina	64 (68.8)
Atypical angina	29 (31.2)
**Medications**	
CCB	19 (20.4)
Beta-blocker	28 (30.1)
ACEI/ARB	17 (18.3)
Statins	31 (33.3)
ASA	22 (23.7)
Clopidogrel	12 (12.9)
**No. of vessels**	103
LAD	84 (81.6)
LCX	19 (18.4)
**Lesion location on vessels**	
Proximal	59 (57.3)
Middle	38 (36.9)
Distal	6 (5.8)
**QCA on per-vessel**	
30% ≤ QCA < 50%	28 (27.2)
50% ≤ QCA < 70%	20 (19.4)
≥70%	55 (53.4)
**Total Agatston score on per vessel**	
<400	79 (76.7)
≥400	24 (23.3)
FFR ≤ 0.80 on per patient	44 (47.3)
FFR ≤ 0.80 on per vessel	47 (45.6)

Data are presented as mean ± SD or number (%), as appropriate. BMI = body mass index; CCB = calcium-channel blockers; ACEI = angiotensin-converting enzyme inhibitors; ARB = angiotensin receptor blocker; ASA = acetylsalicylic acid; LAD = left anterior descending artery; LCX = left circumflex artery; QCA = quantitative coronary angiography; FFR = fractional flow reserve.

**Table 2 jcm-12-02154-t002:** Diagnostic performance of CCTA and CTP-FFR on a per-vessel basis.

	CCTA ≥ 50%	CCTA ≥ 70%	FFR-CT	CTP	CTP-FFR	CTP-FFR + CTP + FFR-CT
Sensitivity	87.2 (74.3–95.2)	74.5 (59.7–86.1)	85.1 (71.7–93.8)	87.2 (74.3–95.2)	87.2 (74.3–95.2)	100.0 (92.5–100.0)
Specificity	53.6 (39.7–67.0)	73.2 (59.7–84.2)	83.9 (71.7–92.4)	75.0 (61.6–85.6)	87.5 (75.9–94.8)	98.2 (90.5–99.9)
Accuracy	68.9 (59.1–77.7)	73.8 (64.2–82.0)	84.5 (76.0–90.9)	80.6 (71.6–87.7)	87.3 (79.4–93.1)	99.0 (94.7–99.9)
PPV	61.2 (53.8–68.1)	70.0 (59.5–78.8)	81.6 (70.7–89.1)	74.6 (64.8–82.4)	85.4 (74.4–92.2)	97.9 (87.1–99.7)
NPV	83.3 (69.5–91.7)	77.4 (67.2–85.1)	87.0 (77.1–93.1)	87.5 (76.6–93.8)	89.1 (79.4–94.6)	100.0 (92.5–100.0)

Values are % (95% CI). PPV = positive predictive value; NPV = negative predictive value. Other abbreviations as in Figure 1.

**Table 3 jcm-12-02154-t003:** Diagnostic performance of CTP-FFR (≤0.80) with different variables in determining myocardial ischemia with invasive fractional flow reserve as a reference standard (≤0.80) on a per-vessel basis.

Analysis Basis	Metrics	Sensitivity	Specificity	Accuracy	PPV	NPV
No. of vessels	LAD	92.7 (80.1–98.5)	93.0 (80.9–98.5)	92.9 (85.1–97.3)	92.7 (80.9–97.4)	93.0 (81.7–97.3)
	LCX	50.0 (11.8–88.2)	69.23 (38.6–90.9)	63.2 (38.4–83.7)	42.9 (19.3–70.2)	75.0 (55.5–87.8)
Lesion location on vessel	Proximal	88.9 (70.8–97.7)	93.8 (79.2–99.2)	91.5 (81.3–97.2)	92.3 (75.7–97.9)	90.9 (77.4–96.7)
	Middle	88.2 (63.6–98.5)	76.2 (52.8–91.8)	81.6 (65.7–92.3)	75.0 (57.8–86.8)	88.9 (68.0–96.8)
	Distal	66.7 (9.43–99.2)	100.0 (29.2–100.0)	83.3 (35.9–99.6)	100.0 (19.8–100.0)	75.0 (37.7–93.7)
QCA on per vessel	30% ≤ QCA < 50%	100.0 (2.5–100.0)	92.6 (75.7–99.1)	92.9 (76.5–99.1)	33.3 (11.6–65.5)	100.0 (83.4–99.9)
	50% ≤ QCA < 70%	100.0 (29.2–100.0)	88.24 (63.6–98.5)	90.0 (68.3–98.8)	60.0 (29.0–84.7)	100.0 (75.9–99.9)
	≥70%	86.0 (72.1–94.7)	75.0 (42.8–94.5)	83.6 (71.2–92.2)	92.5 (82.1–97.1)	60.0 (40.0–77.1)
Total Agatston score on per vessel	<400	89.2 (74.6–97.0)	85.7 (71.5–94.6)	87.3 (78.0–93.8)	84.6 (72.2–92.1)	90.0 (78.0–95.8)
	≥400	80.0 (44.4–97.5)	92.9 (66.1–99.8)	87.5 (67.6–97.3)	88.9 (54.1–98.2)	86.7 (65.1–95.8)

All abbreviations same as in Figure 1, Table 1 and Table 2.

## Data Availability

The data are obtainable on request from the corresponding author in this study. They are not publicly available due to privacy issues.

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
