# Peer review of "A Novel CT Perfusion-Based Fractional Flow Reserve Algorithm for Detecting Coronary Artery Disease"

_jcm, 2023, doi:10.3390/jcm12062154_

Round 1

Reviewer 1 Report

This is an interesting study focused on validation of the newly developed algorithm which calculated CTP-FFR. The aim of the study was to investigate the feasibility and diagnostic performance of this new CTP-FFR in detecting ischemia while using invasive FFR and quantitative coronary angiography (QCA) as reference standards.

Nevertheless, CTP-FFR performed better when compared to QCA, and this is not a correct comparison, as QSA is explicitly anatomical assessment, not physiological. When compared to FFR, then the “grey zone” of the results is quite wide.

Majors                           

1)      How authors correct for the fact that MBP calculated in CT is not a direct MBF, but just the expression of K1? It was previously shown that for CT-MBF a correction is needed to express the real MBF. How was it corrected in the derived CTP-FFR?

2)      What authors mean by „with better agreement with fractional flow reserve 44 (FFR) derived from gold standard invasive coronary angiography” as FFR is not derived from ICA, it is directly measured during this procedure.

3)      Reference 15 provides only the Editor comment, not the study itself.

4)      May Authors specify in Table 2 to which modality all methods were compared? In the text it says it is against FFR, but it is nowhere mentioned in the captions.

5)      It is not clear for the reviewer, why authors compared results against 50% stenosis defined alone by QCA, was it only for left main artery? In the results section there is no clear division between data compared to QCA alone and data compared to QCA+FFR. This should be clarified. Wouldn’t it make sense to validate CTP-FFR explicitly against QCA+FFR? QCA alone does not answers research question, as it does not take into consideration physiology, while CTP-FFR is not an anatomical measurement. This is simply mixing apples with oranges.

6)      Figures entitled as Bland Altman plots does not present the Bland-Altman analysis- or there are errors in the axis’s titles.

7)      Authors underline that CTP-FFR improves detection of lesions 50-69% defined by QCA – however, these lesions are not significant, as data was not verified by FFR. So this still remains in the grey zone

8)      Figure 7 is missing.

9)      When CTP-FFR results were directly compared to FFR, the grey-zone is quite wide: 0.7-0.9, how authors see the improvement of the presented, CTP-FFR, method ?

10)   Authors claims that :” The diagnostic accuracy was higher in the FFR ≤0.7 group than in lesions with an FFR of 0.70 to 0.80 (p=0.014). This suggests that CTP-FFR may achieve excellent agreement in presence of severe ischemia.”. How the abovementioned result may suggest that CTP-FFR may achieve excellent results, as the results show quite the opposite. May you explain that?

Minors

1)      Some sentences are written not in logical order and therefore not understandable, for instance “The CTP-FFR has a strong correlation between CTP-FFR and invasive FFR on per-228 vessel”

Reviewer 2 Report

1.) The paper is need some attention in terms of format and lacks enough review in literature to justify the study of the work.

For example, the paper includes "Traditional FFR-CT based on
computational fluid dynamics (CFD) and deep learning 46 relies on a population-averaged physiological hypothesis model to estimate exit resistance 47 to simulate boundary conditions. However, due to individual differences between pa- 48 tients, this quantification and distribution of total coronary blood flow can affect coronary 49 exit resistance and therefore the accuracy of FFR-CT (9,10)."

This work shouldn't just be limited to coronary exit resistance.  This
can benefit other works including larger arterial models for numerical and experimental works.  The author should add a line or lines of discussion in regard to a review of literature that discusses how this work or other works can benefit numerical and experimental works in cardiovascular flow and even medical drug targeting.  Especially if boundary conditions are the major focus.  There have been several works in which boundary conditions have been simplified partially or fully.

The author includes "Thus, we developed a new algorithm based on the CTP to measure FFR through out- 64 lining myocardial perfusion regions and improving boundary conditions"  This further justifies the need for more referencing.

Publications to include:

Shi, Y., Peng, C., Liu, J.Z., et al, “A Modified Method of Computed Fluid Dynamics Simulation in Abdominal Aorta and Visceral Arteries”, Computer Methods in Biomechanics and Biomedical Engineering, vol 25, issue 15, pp 1718-1729, 2021

Stanley, N., Timms, W., Ciero, A., and Hewlin, R.L., “Development of 3-D Printed Optically Clear Rigid and Anatomical Vessels for Particle Image Velocimetry Analysis in Cardiovascular Flow", ASME International Mechanical Engineering Congress and Exposition, Salt Lake City, Utah, November 11-14, 2019

Sun, Z.H., and Xu, L. “Computational Fluid Dynamics in Coronary Artery Disease”, Computerized Medical Imaging and Graphics, Vol 38, 8, pp. 651-663

Kambayashi, Y., Takao, H., et al. “Computational fluid dynamics analysis of tandem carotid artery stenoses: Investigation of neurological complications after carotid artery stenting”, Technology and Health Care, vol 24, issue 5, pp. 673-670

Jhunjhunwala, P, Padole, P.M., and Thombre, S.B., “CFD Analysis of Pulsatile Flow and Non-Newtonian Behavior of Blood in Arteries”, Molecular & Cellular Biomechanics, vol, 12, issue 1, pp 37-47, 2015

Hewlin, R.L. and Kizito, J.P., “Evaluation of the Effect of Simplified
and Patient-Specific Arterial Geometry on Hemodynamic Flow in Stenosed Carotid Bifurcation Arteries”, ASME Early Technical Career Conference, Atlanta, Georgia, November 4-5, 2011

The following  recent works (2019-2023)  will be helpful for a citation list of works in cardiovascular medical drug delivery which also include boundary condition modelling:

Karvelas, E.G., Lampropoulos, N.K., Karakasidis, T.E., et al. “Blood Flow and Diameter Effect in the Navigation Process of Magnetic Nanocarriers inside the Carotid Artery”, Computer Methods and Programs in Biomedicine, vol 221, 2022

Karvela, E., Liosis, C., et al, “An Optimized Method for 3D Magnetic Navifation of Nanoparticles inside Human Arteries”, Fluids, vol 6, issue 3, 2021

2.) Figures should be enlarged to show the full image and scales of
color bars.

3.) Some of the discussion needs to be moved to the conclusion sections.  A major part of the conclusive remarks are found in the discussion.  I suggest moving this to the conclusion.

4.) Equations should be numbered and referenced.

5.) In the statistical analysis section, data needs to be presented and referenced to.  There is a discussion but no visual data.

6.) There are a number of hanging figures.  For example Figure 2 is shown, but not referenced in the text.  Please address this for all figures.

Round 2

Reviewer 2 Report

1.) For the previous comment on the paper is need some attention in terms of format and lacks enough review in literature to justify the study of the work, please include all of the citations from the previous list of suggested citations. This will provide a more in-depth review of prior works related to experimental and computational works related to cardiovascular flow and boundary condition assignment.
